# Concentration-Dependent Attenuation of Pro-Fibrotic Responses after Cannabigerol Exposure in Primary Rat Hepatocytes Cultured in Palmitate and Fructose Media

**DOI:** 10.3390/cells12182243

**Published:** 2023-09-09

**Authors:** Klaudia Sztolsztener, Karolina Konstantynowicz-Nowicka, Anna Pędzińska-Betiuk, Adrian Chabowski

**Affiliations:** 1Department of Physiology, Medical University of Bialystok, 15-089 Bialystok, Poland; karolina.konstantynowicz-nowicka@umb.edu.pl (K.K.-N.); adrian.chabowski@umb.edu.pl (A.C.); 2Department of Experimental Physiology and Pathophysiology, Medical University of Bialystok, 15-089 Bialystok, Poland; anna.pedzinska-betiuk@umb.edu.pl

**Keywords:** cannabigerol, hepatocytes, fibrosis, extracellular matrix, collagen

## Abstract

Hepatic fibrosis is a consequence of liver injuries, in which the overproduction and progressive accumulation of extracellular matrix (ECM) components with the simultaneous failure of matrix turnover mechanisms are observed. The aim of this study was to investigate the concentration-dependent influence of cannabigerol (CBG, *Cannabis sativa L.* component) on ECM composition with respect to transforming growth factor beta 1 (TGF-β1) changes in primary hepatocytes with fibrotic changes induced by palmitate and fructose media. Cells were isolated from male Wistar rats’ livers in accordance with the two-step collagenase perfusion technique. This was followed by hepatocytes incubation with the presence or absence of palmitate with fructose and/or cannabigerol (at concentrations of 1, 5, 10, 15, 25, 30 µM) for 48 h. The expression of ECM mRNA genes and proteins was determined using PCR and Western blot, respectively, whereas media ECM level was evaluated using ELISA. Our results indicated that selected low concentrations of CBG caused a reduction in TGF-β1 mRNA expression and secretion into media. Hepatocyte exposure to cannabigerol at low concentrations attenuated collagen 1 and 3 deposition. The protein and/or mRNA expressions and MMP-2 and MMP-9 secretion were augmented using CBG. Considering the mentioned results, low concentrations of cannabigerol treatment might expedite fibrosis regression and promote regeneration.

## 1. Introduction

Hepatic fibrosis is a considerable pathological consequence of persistent liver injuries, such as hepatitis virus infection, steatosis, and other metabolic disorders, which can lead to a gradual loss of liver function. It can be defined as an excessive scar tissue formation due to the overproduction and progressive accumulation of extracellular matrix (ECM) components with the simultaneous failure of matrix turnover mechanisms, leading to architectural changes and finally contributing to hepatocytes’ dysfunction. This may be an early signal for the occurrence of cirrhosis and, ultimately, organ failure [1,2]. The current data suggest that the development of hepatic fibrosis processes is orchestrated by parenchymal and mesenchymal cells, releasing a complex network of pro-fibrogenic cytokines, and predominantly transforming growth factor beta 1 (TGF-β1), which is involved in cells’ activation and differentiation into a heterogeneous population of myofibroblasts [2]. The activation of the TGF-β1 signaling pathway is a fundamental to liver fibrogenesis, in which the components of the extracellular matrix are extensively remodeled [3]. TGF-β1 activation stimulates the transcription of procollagen type 1 and 3 genes, favoring matrix production and the deposition of collagen (COL) [2]. Altering the homeostasis of ECM macromolecules, which are most commonly found in fibrous proteins, i.e., collagen, promotes the generation of the fibril network, which gradually replaces the normally low-density matrix of the basement membrane, the liver parenchyma [2,4]. Additionally, the deposition of other matrix molecules, such as fibronectin or elastin, occurs, thereby stabilizing the structure of fibril-forming collagen [2]. ECM homeostasis depends on the balance between the expression of matrix metalloproteinases (MMP) and their tissue inhibitors, the tissue inhibitors of matrix metalloproteinases (TIMP) [3,5]. During fibrogenesis, the expression of MMP and TIMP is upregulated, while the enhanced production of TIMP additionally inhibits MMP activity, preventing MMP-dependent matrix protein degradation in the extracellular space [3,6,7]. Different fibrogenic cell types take part in the ECM remodeling process, such as hepatic stellate cells (HSC), hepatocytes, and Kupffer cells [2,5,8]. It is accepted that hepatocytes play an important role in cellular differentiation, proliferation, and ECM synthesis and degradation. The TGF-β1 cytokine secreted by hepatocytes activates the intracellular signal transduction SMAD pathway, which directly regulates the transcription of target genes (e.g., COL, MMP, and TIMP) [3,8]. This seems to be a critical in protecting the fibrotic matrix’s remodeling during liver injury. Therefore, the inhibition of the TGF-β1 pathway may be a target for the experimental analysis of an agent with potential anti-fibrotic activity. At present, the non-psychotropic component of medical marijuana (*Cannabis sativa* L.), namely, cannabigerol (CBG), appears to be a potential therapeutic agent for treating a variety of medical conditions. Studies indicate that CBG reduces the formation of reactive oxygen species in the intestinal epithelial cells during irritable bowel syndrome and also has potential anti-inflammatory effects that are similar to other cannabinoids [9,10]. Importantly, CBG acting on various receptors in the endocannabinoidome (eCBome) has an impact on the regulation of inflammation as well as processes related to lipid metabolism and overall hepatic metabolic functioning [11]. It appears that cannabigerol, which is a partial agonist of cannabinoid receptor 2 (CB_2_), may lead to a reduction in the liver injury and accelerate liver regeneration [12]. The investigation of liver biopsy samples from patients with fibrosis revealed that these tissues were characterized by up-regulated CB_2_ expression, which is not detected in healthy liver [13]. In order to better understand the therapeutic potential of CBG, we will examine the detailed influence of this phytocannabinoid on the ECM composition of hepatocytes exposed to palmitate and fructose. In this study, we will focus on CBG’s effects on the first step of fibrosis in primary rat hepatocytes, notably the TGF-β1 signaling pathway, as a key factor in the development of hepatic fibrotic changes.

## 2. Materials and Methods

### 2.1. Animals and Liver Perfusion Procedure

This experiment was conducted with the permission of the Local Ethical Committee for Animal Experiments in Olsztyn (No. 79/2022). Male Wistar rats with an initial body weight of approximately 150–200 g were maintained in standard animal-holding conditions, as follows: 22 ± 2 °C air temperature on reversed light/dark cycle (12 h/12 h) with ad libitum access to water and standard rodent chow (Labofeed B diet; energy distribution of kcal: 67% carbohydrates, 25% protein, 8% fat; Kcynia, Poland). After one week of acclimatization, the rats were anesthetized by intraperitoneal pentobarbital injection (in a dosage of 80 mg/kg of body mass) and liver perfusion was performed in accordance with the two-step perfusion technique (with collagenase and ethylenedinitrilotetraacetic acid (EDTA)), as previously described by Seglen [14]. Firstly, the hepatic portal vein was cannulated with Hank’s Balanced Salt Solution (HBSS, without Ca^2+^ and Mg^2+^; Sigma Aldrich, Saint Louis, MO, USA), supplemented with 2 mM EDTA (Sigma Aldrich, Saint Louis, MO, USA). Next, the tissue was perfused with HBSS (without Ca^2+^ and Mg^2+^; Sigma Aldrich, Saint Louis, MO, USA) and supplemented with 0.05% collagenase (Sigma Aldrich, Saint Louis, MO, USA), until the digestion intracellular junctions and the destruction of hepatic structure occurred. Following that, the liver was gently dissected and cells were released and dispersed into culture-plating media (87% Dulbecco Modified Eagle Medium (DMEM) with 4.5 g/L glucose (PAN-Biotech, Aidenbach, Germany), 10% fetal bovine serum (FBS; PAN-Biotech, Aidenbach, Germany), 1% antibiotic/antimycotic (penicillin/streptomycin; Biowest, Nuaillé, France), and 1% N-2-Hydroxyethylpiperazine-N′-2-ethanesulfonic acid (HEPES; Gibco; Thermo Fisher Scientific, Inc., Waltham, MA, USA), 1% EDTA (Sigma Aldrich, Saint Louis, MO, USA)). The cell suspension consisting of mixed hepatic cells and debris was filtered through a 100µm cell strainer. The filtrate was centrifuged at 50× *g* for 5 min at 4 °C and washed three times with culture-plating media. Then, hepatocytes were density-separated in 90% Percoll solution (Sigma Aldrich, Saint Louis, MO, USA) during centrifugation at 200× *g* for 10 min at 4 °C. The obtained hepatocyte precipitates were washed once with DMEM-plating media [15]. The hepatocytes’ concentration, size distribution, and viability were immediately determined using Trypan blue staining on the LUNA automated cell-counter (Logos Biosystems, Aligned Genetics, Inc., Annandale, VA, USA). The prepared mixture of 10% cell suspension in Phosphate-Buffered Saline (PBS; PAN-Biotech, Aidenbach, Germany), and 0.4% Trypan blue (1:1, *v*/*v*) was applied to a cell-counting slide chamber and left until the result was obtained. The experiment was conducted using 10^6^ cells per well, with a viability above 85%.

### 2.2. Preparation of Reagents for Cell Treatment

Prior to cell treatment, the stock solution of palmitate (PA), fructose (F), and cannabigerol (CBG) was prepared as described below. Palmitic acid (Sigma Aldrich, Saint Louis, MO, USA) was dissolved in a mixture of 1M NaOH and absolute ethanol by heating at 70 °C until a homogenous solution was obtained, and then conjugated with fatty acid-free bovine serum albumin (BSA; Sigma Aldrich, Saint Louis, MO, USA) at a final concentration of 8 mM palmitate and 10% BSA. Fructose (Sigma Aldrich, Saint Louis, MO, USA) was dissolved at a concentration of 300 mM in DMEM. A concentrated (3 mM) stock solution of cannabigerol (TargetMol, Boston, MA, USA) was prepared in dimethyl sulfoxide (DMSO; Sigma Aldrich, Saint Louis, MO, USA).

### 2.3. Cell Treatment

Following isolation, primary rat hepatocytes were seeded on Petri dishes and cultured in standard cell culture medium consisting of 89% DMEM with 4.5 g/L glucose (PAN-Biotech, Aidenbach, Germany), 10% FBS (PAN-Biotech, Aidenbach, Germany), and 1% antibiotic/antimycotic (Biowest, Nuaillé, France). After 1h of hepatocytes’ relaxation, the media were removed and cells were flooded a fresh experimental medium: (1) Control group, with cells cultured in standard media for 48 h; (2) Palmitic Acid with Fructose (PA-F) group, with cells exposed to 0.4 mM palmitate in combination with 10 mM fructose for 48 h. Among the Control and PA-F groups, half of the primary hepatocytes were incubated with 1, 5, 10, 15, 25, or 30 μM cannabigerol for 48 h (the CBG concentrations were selected based on previous data) [16,17]. The concentration of palmitate and fructose was selected based on the available literature indicating the promotion of the hepatic fibrosis pathway through the activation of pro-fibrotic gene expression [7,8,18]. The media were replaced with fresh media after 24 h incubation. At the end of the experiment, cell morphology was assessed in the Bürker chamber using Trypan blue-staining for a qualitative estimation of living cells. The cells with medium were transferred from Petri dishes to falcons and centrifuged at 50× *g* for 6min at 4 °C. The media samples after 24 h and 48 h incubation were collected and immediately frozen at the temperature of liquid nitrogen. The cells were washed two times with ice-cold PBS (PAN-Biotech, Aidenbach, Germany) and centrifuged. After that, hepatocytes were suspended in an ice-cold radioimmunoprecipitation assay (RIPA) lysis buffer containing protease and phosphatase inhibitors (Roche Diagnostics, Boston, MA, USA), or in ice-cold PBS (PAN-Biotech, Aidenbach, Germany), or lysed in Trizol reagent (Sigma Aldrich, Saint Louis, MO, USA). Medium and cell lysates were stored at −80 °C until further determination.

The entire range of cannabigerol concentrations (1–30 µM) selected for this study was used to perform the experiment within the same rat. This experiment was repeated on four independent animals to eliminate the influence of individual rats’ characteristics and changing experimental conditions. In the experiments conducted on the same one animal, the number of independent replicates was set to 6 (*n* = 4 rats; 6 replicates per each experimental group from the same animal).

### 2.4. Cell Cytotoxicity

For the assessment of cellular cytotoxicity, the lactate dehydrogenase (LDH) concentration was determined. The quantitative detection of LDH released into a culture medium was performed using a rat lactate dehydrogenase enzyme-linked immunoassay (ELISA) kit from Enlibio Biotech Co., Ltd. (Wuhan, China) based on a double-antibody sandwich technique.

In brief, the cell culture supernatant was centrifuged at 1000× *g* for 15 min at 4 °C and then added to an appropriate well of pre-coated antibody (anti-Rat LDH monoclonal antibody) plate. The standard dilutions were also added to an appropriate well of plate. After incubation and washing, the biotinylated antibody was inserted in a 96-well plate, after which the plate was incubated again. Then, enzyme conjugate was added and the plate was incubated and rinsed. The color reagent was added, followed by incubation. Finally, color reagent C, as a stop solution, was pipetted into each well and the absorbance was read at 450 nm on a microplate Synergy H1 Hybrid Reader (BioTek Instruments, Winooski, VT, USA). The LDH concentration was calculated relative to the standard curve and expressed as a percentage of LDH released relative to the Control (set as 100%).

### 2.5. Protein Extraction and Western Blot Assay

The primary rat hepatocytes were homogenized in ice-cold RIPA buffer containing a cocktail of protease and phosphate inhibitors. Then, all samples were centrifuged at 10,000× *g* for 30 min at 4 °C and supernatants were collected into new tubes. In the obtained supernatant fraction, the protein concentration was determined using the bicinchoninic (BCA) method with bovine serum albumin as a standard. Following that, all samples were diluted in Laemmli buffer (Bio-Rad Laboratories, Inc., Hercules, CA, USA) to the same protein concentration (30 µg), loaded on CriterionTM TGX Stain-Free Precast Gels (Bio-Rad Laboratories, Inc., Hercules, CA, USA), and separated by sodium dodecyl sulfate–polyacrylamide gel electrophoresis (SDS-PAGE). After electrophoresis, proteins were transferred onto nitrocellulose or polyvinylidene fluoride (PVDF) membranes for wet or semi-dry transfer systems, respectively. The membranes were incubated with 5% non-fat dry milk or BSA in tris-buffered saline with the addition of Tween-20 (TBST), which prevents the non-specific binding of antibodies to the blotting membrane. The membranes were immunoblotted overnight with primary antibodies, i.e., transforming growth factor beta 1 (TGF-β1), metalloproteinase 2 (MMP-2), metalloproteinase 9 (MMP-9), tissue inhibitor of metalloproteinase 1 (TIMP-1), tissue inhibitor of metalloproteinase 2 (TIMP-2), collagen type 1 alpha 1 (COL-1α1), and collagen type 3 alpha 1 (COL-3α1). The details of primary antibodies and dilutions are shown in Table 1. The next day, the protein-blotting membranes were immunoblotted with horseradish peroxidase (HRP)-linked secondary antibodies and, finally, the chemiluminescent visualization of the protein bands was achieved using Clarity Western ECL substrate (Bio-Rad, Hercules, CA, USA). The obtained protein signals were densitometrically quantified using the ChemiDoc visualization system (Image Lab Software; Bio-Rad, Warsaw, Poland). The expression of target proteins was standardized to the expression of total protein as a control, which was set as 100%.

### 2.6. RNA Extraction and Quantitative Real-Time PCR Assay

Total RNA was extracted from primary rat hepatocytes by the TriReagent RNA isolation technique (Sigma Aldrich, Saint Louis, MO, USA) according to the manufacturer’s instruction. The quantity and quality of obtained RNA were spectrophotometrically determined at the absorbance O.D. (260/280 nm ratio) on a microplate Synergy H1 Hybrid Reader (BioTek Instruments, Winooski, VT, USA). After RNA isolation, the first-strand cDNA was synthesized from 1 µg of total RNA by the EvoScript Universal cDNA Master kit (Roche Molecular Systems, Boston, MA, USA) under the following incubation conditions: 42 °C for 900 s, 85 °C for 300 s, 65 °C for 900 s, and 4 °C for at least 240 s. The quantitative Real-Time Polymerase Chain Reaction (qRT-PCR) was conducted using FastStart Essential DNA Green Master kit (Roche Molecular Systems, Boston, MA, USA) in accordance with the following conditions: preincubation, 3-step amplification (denaturation at 95 °C for 15 s, annealing (temperature of selected primers is shown in Table 2) for 15 s, extension at 72 °C for 15 s), melting. The synthesis of the cDNA and qRT-PCR procedures was performed on LightCycler 96 Instrument with real-time thermal cycler (Roche Diagnostics, Boston, MA, USA). The specificity of the obtained PCR product was confirmed by analysis of the melting curve. All reactions were carried out in duplicate. The gene expression levels were normalized to the housekeeper gene, i.e., glyceraldehyde-3-phosphate dehydrogenase (GAPDH), and calculated in accordance with the relative quantification method modified by Pfaffl [19]. The mRNA expression of target genes in the Control group was set as 1. The sequence of selected primers is listed in Table 2.

### 2.7. Immunoassay

For the assessment of TGF-β1, MMP-2, MMP-9, TIMP-1, and TIMP-2 contents in cell culture supernatants, the commercially available ELISA kits were used (for TGF-β1, TIMP-1, and TIMP-2 from Thermo Fisher Scientific, Inc., Waltham, MA, USA; for MMP-2 and MMP-9 from Boster Biological Technology, Ltd., Pleasanton, CA, USA). Before determination, the cell culture supernatants were centrifuged at 1000× *g* for 15 min at 4 °C (for MMP-2, MMP-9, TIMP-1, and TIMP-2 measurements) or used for TGF-β1 protein extraction by adding the solutions of the first 1M HCl and second 1M NaOH. Then, the samples and standard dilutions were added to a proper well of pre-coated antibody plates and incubated. After the addition of a Biotinylated proper antibody and washing, as well as the addition of Streptavidin-HRP and subsequent rinses, the TMB substrate was added, followed by further incubation. Finally, the Stop Solution was pipetted into each well and the absorbance was read at 450 nm on a microplate Synergy H1 Hybrid Reader (BioTek Instruments, Winooski, VT, USA). The concentration of measured parameters was calculated relative to the appropriate standard curve and expressed as picogram per milliliter of media (pg/mL).

The hydroxyproline (HYP) concentration in cell lysates was determined using the competitive enzyme immunoassay kit obtained from BlueGene Biotech Co., Ltd. (Shanghai, China). Before measurement, cells resuspended in PBS were ultrasonicated three times and then centrifuged at 1000× *g* for 15 min at 4 °C to obtain cell lysates purified of cell debris contamination. The samples and standard dilutions were added to an appropriate pre-coated antibody (anti-rat HYP monoclonal antibody) plate. After the addition of Balance Solution and Enzyme conjugate to 96 wells, the plate was incubated. Next, the plate was washed 5 times and the mixtures of Substrate A and Substrate B was added, followed by further incubation. Finally, the Stop Solution was pipetted into each well and the absorbance was read at 450nm on a microplate Synergy H1 Hybrid Reader (BioTek Instruments, Winooski, VT, USA). The concentration of HYP was calculated, corresponding to the standard curve, and expressed as microgram per milliliter of cell lysates (µg/mL).

### 2.8. Statistical Analysis

The data from the experiment were expressed as mean ± standard deviation (SD) based on six independent determinations (*n* = 6). Statistical analyses were carried out using GraphPad Prism version 8.2.1. (GraphPad Software, San Diego, CA, USA). The normal distribution of values and the homogeneity of variances were verified using the Shapiro–Wilk and Bartlett’s tests, respectively. For values with normal distribution, the statistical comparisons were assessed by the parametric *t*-test; for values with abnormal distribution, comparisons were assessed using the non-parametric Mann–Whitney U test. The statistical differentiation between experimental groups were also checked using two-way ANOVA supported by Tukey’s test. The statistical significance level is considered *p* < 0.05.

## 3. Results

### 3.1. Concentration-Dependent Influence of Cannabigerol on Cytotoxicity in Post-Incubation Media from Primary Rat Hepatocytes Exposed to Palmitate in Combination with Fructose

In the cell culture media, after 24 h treatment of hepatocytes with 1 µM CBG, the release of LDH was significantly enhanced (+0.9%, *p* < 0.05; Figure 1A) compared with the Control group. Moreover, the LDH release into media was decreased in the group of hepatocytes exposed to PA-F with 1 µM CBG (−1.7%, *p* < 0.05; Figure 1A) and increased in the group of hepatocytes exposed to PA-F with 10 µM CBG (+3.4%, *p* < 0.05; Figure 1A) compared with the PA-F group. We also observed a decline in the level of LDH release to cell culture supernatant after hepatocyte treatment (24 h) with palmitate and fructose media with 15 µM CBG, and palmitate and fructose media with 25 µM CBG (−3.4% and −3.4%, respectively, *p* < 0.05; Figure 1B), in relation to the Control group. At the end of our experiment, the released LDH content was higher in the culture supernatant from hepatocytes treated by standard media with the addition of 5 µM CBG and 10 µM CBG (+3.9% and +2.9%, respectively, *p* < 0.05; Figure 1C) than it was in the Control group. The significant alterations in LDH release were also noted in the PA-F + 5 µM CBG group (−1.5%, *p* < 0.05; Figure 1D) compared with the PA-F group.

### 3.2. Concentration-Dependent Influence of Cannabigerol on Transforming Growth Factor Beta 1 Changes in Post-Incubation Media and Primary Rat Hepatocytes Exposed to Palmitate in Combination with Fructose

In the cell culture media, the concentration of TGF-β1 was enlarged after hepatocytes underwent 24 h incubation with palmitate and fructose (+14.3% and +18.4%, respectively, *p* < 0.05; Figure 2A,B). The 24 h cell treatment with 5 µM CBG in combination with PA and F induced a decrease in the TGF-β1 secretion to media (−21.8%, *p* < 0.05; Figure 2A) compared with that in the PA-F group. We also observed an increase in the TGF-β1 concentration after 24 h incubation in the PA-F + 10 µM CBG group (+15.1%, *p* < 0.05; Figure 2A) in relation to the Control group. The secretion of TGF-β1 into media (24 h) was also increased in the following groups: 15 µM CBG, 25 µM CBG, 30 µM CBG (+21.0%, +59.7% and +59.9%, respectively, *p* < 0.05; Figure 2B) compared with the standard condition. The cells’ incubation with PA-F and 30 µM CBG for 24 h enhanced the concentration of TGF-β1 (+39.2% and +17.6%, *p* < 0.05; Figure 2B) compared with the Control and PA-F groups, respectively. At the end of the experiment, the TGF-β1 level in post-incubation media was elevated in the PA-F + 10 µM CBG group (+38.0% and +29.8%, *p* < 0.05; Figure 2C; vs. Control and PA-F groups, respectively) and PA-F + 30 µM CBG group (+7.7%, *p* < 0.05; Figure 2D; vs. Control group). An increase in the TGF-β1 level was also noted at the end of cells’ exposure to 30 µM CBG (+43.5%, *p* < 0.05; Figure 2D) in comparison with the standard condition.

The protein and mRNA expressions of TGF-β1 are presented in Appendix A in the Appendix A.

### 3.3. Concentration-Dependent Influence of Cannabigerol on the Hydroxyproline Content in Primary Rat Hepatocytes Exposed to Palmitate in Combination with Fructose

The content of hydroxyproline was increased in both PA-F groups (+159.1% and +179.5%, *p* < 0.05; Figure 3A,B, respectively). The treatment with 1 µM CBG, 5 µM CBG, and 10 µM CBG alone caused significant alterations in HYP content (−61.2%, +71.2% and −46.1%, *p* < 0.05; Figure 3A, respectively). The level of HYP was enlarged in all examined groups, in which hepatocytes were incubated with high concentrations of CBG under standard or PA-F conditions (15 µM CBG: +134.7%, 25 µM CBG: +194.3%, 30 µM CBG: +271.0%, PA-F + 15 µM CBG: +460.1%, PA-F + 25 µM CBG: +327.9%, PA-F + 30 µM CBG: +737.7%, *p* < 0.05; Figure 3B) in comparison with the Control group. In relation to the PA-F group, hydroxyproline concentration was enhanced in the lysates of hepatocytes incubated with PA-F and selected concentrations of CBG (PA-F + 15 µM CBG: +100.4%, PA-F + 25 µM CBG: +53.1%, PA-F + 30 µM CBG: +199.7%, *p* < 0.05; Figure 3B).

### 3.4. Concentration-Dependent Influence of Cannabigerol on Collagen Type 1 Alpha 1 Changes in Primary Rat Hepatocytes Exposed to Palmitate in Combination with Fructose

The protein expression of COL-1α1 was enhanced in both PA-F groups (+55.2% and +41.2%, *p* < 0.05; Figure 4A,B, respectively). In comparison with the Control group, the protein expression of COL-1α1 was decreased in the 10 µM CBG, and PA-F + 5 µM CBG groups (−19.5%, and −30.4%, *p* < 0.05; Figure 4A, respectively) and increased in the PA-F + 1 µM CBG group (+22.5%, *p* < 0.05; Figure 4A). We also observed a diminishment in the COL-1α1 protein expression after treatment with low concentrations of CBG in the PA-F condition (PA-F + 1 µM CBG: −19.5%, PA-F + 5 µM CBG: +54.3%, PA-F + 10 µM CBG: −46.9%, *p* < 0.05; Figure 4A) compared with the PA-F group. Moreover, an increment in the COL-1α1 protein expression was detected after CBG treatment in the following groups: 15 µM CBG, PA-F + 15 µM CBG, PA-F + 25 µM CBG, PA-F + 30 µM CBG (+29.0%, +19.5%, +23.8%, +22.0%, *p* < 0.05; Figure 4B, respectively) in relation to the Control group.

The mRNA expression of COL-1α1 was increased in both PA-F groups (+22.1% and +24.8%, *p* < 0.05; Figure 4C,D, respectively). We also observed an increase in the COL-1α1 mRNA expression in the 1 µM CBG, PA-F + 10 µM CBG, and PA-F + 15 µM CBG groups (+39.4%, +27.7%, +31.6%, *p* < 0.05; Figure 4C,D, respectively) in relation to the proper Control group. In addition, the mRNA expression of COL-1α1 was reduced after treating hepatocytes with 5 µM CBG, 25 µM CBG, and 30 µM CBG in the PA-F condition (−21.6%, −18.8%, −52.3%, *p* < 0.05; Figure 4C,D, respectively) compared with the appropriate PA-F group. Compared with the standard condition, the mRNA expression of COL-1α1 was decreased in the PA-F + 30 µM CBG group (−40.5%, *p* < 0.05; Figure 4D).

### 3.5. Concentration-Dependent Influence of Cannabigerol on Collagen Type 3 Alpha 1 Changes in Primary Rat Hepatocytes Exposed to Palmitate in Combination with Fructose

In the case of COL-3α1 protein expression, an elevation was observed in both PA-F groups (+20.0% and +12.0%, *p* < 0.05; Figure 5A,B, respectively). The 1 µM CBG and 25 µM CBG treatments of hepatocytes exposed to palmitate and fructose caused a decrease in the COL-3α1 protein expression (−10.7% and −15.2%, *p* < 0.05; Figure 5A,B, respectively) compared with the proper Control group. In all groups of hepatocytes exposed to PA-F and CBG, except the PA-F + 15 µM CBG group, the COL-3α1 protein expression was lowered (PA-F + 1 µM CBG: −25.6%, PA-F + 5 µM CBG: −21.9%, PA-F + 10 µM CBG: −20.4%, PA-F + 25 µM CBG: −24.3%, PA-F + 30 µM CBG: −24.0%, *p* < 0.05; Figure 5A,B) compared with the proper PA-F group.

The mRNA expression of COL-3α1 was increased in both PA-F groups (+110.6% and +76.2%, *p* < 0.05; Figure 5C,D, respectively). In comparison with the proper PA-F group, we observed a diminishment in COL-3α1 mRNA expression after hepatocyte treatment with low concentrations of CBG under PA-F conditions (PA-F + 1 µM CBG: −48.0%, PA-F + 5 µM CBG: −35.5%, PA-F + 10 µM CBG: −43.2%, *p* < 0.05; Figure 5C). In the PA-F + 5 µM CBG group, the COL-3α1 mRNA expression was higher (+35.9%, *p* < 0.05; Figure 5C) than in the Control group. Our study also demonstrated an increase in the COL-3α1 mRNA expression after high concentrations of CBG treatment in cells exposed to palmitate and fructose media (PA-F + 15 µM CBG: +97.4%, PA-F + 25 µM CBG: +101.4%, PA-F + 30 µM CBG: +75.5%, *p* < 0.05; Figure 5D) in comparison with the Control group.

### 3.6. Concentration-Dependent Influence of Cannabigerol on Matrix Metalloproteinase 2 Changes in Post-Incubation Media and Primary Rat Hepatocytes Exposed to Palmitate in Combination with Fructose

In the culture media, the concentration of MMP-2 was decreased after 24 h hepatocyte incubation with palmitate and fructose media (−21.0% and −20.5%, *p* < 0.05; Figure 6A,B, respectively). The secretion of MMP-2 into media was reduced in the following groups: 5 µM CBG, 10 µM CBG, PA-F + 10 µM CBG, PA-F + 25 µM CBG, and PA-F + 30 µM CBG as well (−23.1%, −23.0%, −24.9%, −26.6%, and −8.6%, *p* < 0.05; Figure 6A,B, respectively) compared with the proper Control group. The 24 h cell treatment with PA-F in combination with 25 µM CBG decreased, whereas the treatment in combination with 30 µM CBG increased, the MMP-2 secretion to media (−7.6% and +15.0%, *p* < 0.05; Figure 6B) compared with the PA-F condition. At the end of our experiment, we observed a reduction in the MMP-2 concentration after incubation with 1 µM CBG in hepatocytes exposed to the standard and PA-F conditions (−48.0% and −28.1%, *p* < 0.05; Figure 6C, respectively) in relation to the Control group. At the end of the experiment, the MMP-2 levels in post-incubation media were elevated in the PA-F + 5 µM CBG group (+22.8% and +24.4%, *p* < 0.05; Figure 6C, vs. Control and PA-F groups, respectively) and declined in the PA-F + 15 µM CBG group (−40.0% and −41.9%, *p* < 0.05; Figure 6D, vs. Control and PA-F groups, respectively). A diminishment in the MMP-2 level was also noted at the end of cells’ exposure to the 25 µM CBG and PA-F media (−15.0%, *p* < 0.05; Figure 6D) in comparison with the PA-F condition.

The protein and mRNA expressions of MMP-2 are presented in Appendix A, Appendix A.

### 3.7. Concentration-Dependent Influence of Cannabigerol on Matrix Metalloproteinase 9 Changes in Post-Incubation Media and Primary Rat Hepatocytes Exposed to Palmitate in Combination with Fructose

In the culture media, the concentration of MMP-9 was higher after 24 h cell treatment with 1 µM CBG, 5 µM CBG, 15 µM CBG, and 25 µM CBG under the standard condition (+20.2%, +23.4%, +23.4%, and +46.5%, *p* < 0.05; Figure 7A,B, respectively). The 24 h cell treatment with 30 µM CBG alone induced a decrease in MMP-9 release into media (−38.7%, *p* < 0.05; Figure 7B, vs. Control group). The secretion of MMP-9 into media (after 24 h incubation) was also enhanced in the following groups: PA-F + 1 µM CBG, PA-F + 5 µM CBG, PA-F + 10 µM CBG, PA-F + 15 µM CBG, and PA-F + 30 µM CBG (+26.1%, +58.6%, +26.1%, +18.6% and +27.8%, *p* < 0.05; Figure 7A,B, respectively) compared with the proper PA-F group. We also noted an increase in the MMP-9 level in media after 24 h treatment with 5 µM CBG with the addition of PA-F (+30.7%, *p* < 0.05; Figure 7A) compared with the Control group. In the culture media, the concentration of MMP-9 was reduced at the end of our experiment in both PA-F groups (−11.7% and −17.3%, *p* < 0.05; Figure 7C,D, respectively). At the end of our experiment, we also observed a diminishment in the MMP-9 concentration in almost all CBG-treated groups (1 µM CBG: −9.0%, PA-F + 1 µM CBG: −13.8%, PA-F + 10 µM CBG: −19.4%, 15 µM CBG: −16.8%, 25 µM CBG: −40.7%, 30 µM CBG: −16.8%, PA-F + 15 µM CBG: −66.2%, PA-F + 30 µM CBG: −24.7%, *p* < 0.05; Figure 7C,D) in comparison with the appropriate Control group. In the PA-F group, hepatocyte treatment (48 h) with 5 µM CBG increased and hepatocyte treatment (48 h) with 15 µM CBG decreased the MMP-9 secretion into media (+34.7% and −59.1%, *p* < 0.05; Figure 7C,D, respectively) in comparison with the PA-F condition.

The protein and mRNA expressions of MMP-9 are presented in Appendix A, Appendix A.

### 3.8. Concentration-Dependent Influence of Cannabigerol on Tissue Inhibitor of Metalloproteinase 1 Changes in Post-Incubation Media and Primary Rat Hepatocytes Exposed to Palmitate in Combination with Fructose

Both 24 h incubation with PA and F caused an increase in the TIMP-1 secretion into media (+25.1% and +25.5%, *p* < 0.05; Figure 8A,B, respectively). In the culture media, the concentration of TIMP-1 was also higher after 24 h cell treatment with 1 µM CBG, 15 µM CBG, and 30 µM CBG in the standard condition (+29.8%, +17.9%, and +45.6%, *p* < 0.05; Figure 8A,B, respectively). The release of TIMP-1 into media was changed in almost all CBG with PA-F-treated groups (PA-F + 1 µM CBG: +58.7% and +26.8%, PA-F + 10 µM CBG: +86.8% and +49.3%, PA-F + 15 µM CBG: −25.5% and −40.6%, PA-F + 30 µM CBG: −57.8% and −66.3%, *p* < 0.05; Figure 8A,B) in comparison with the appropriate Control and PA-F groups, respectively. The 24 h cell treatment with 5 µM CBG alone induced a decrease in the TIMP-1 content in post-incubation media (−17.7%, *p* < 0.05; Figure 8A, vs. Control group). We also noted an increase in the TIMP-1 level in media after 24 h cell treatment with PA-F and the addition of 25 µM CBG (+27.5%, *p* < 0.05; Figure 8B) compared with the Control group. At the end of our experiment, we also observed an enhancement in the TIMP-1 concentration in hepatocytes treated with 1 µM CBG alone and 25 µM CBG alone (+20.8% and +6.1%, *p* < 0.05; Figure 8C,D, respectively) in comparison with the appropriate Control group. Moreover, at the end of the experimental incubation with 10 µM CBG, we observed a decrease in TIMP-1 secretion into media (10 µM CBG: −4.8%, PA-F + 10 µM CBG: −21.1%, *p* < 0.05; Figure 8C, vs. Control group). In comparison with the PA-F condition, at the end of the experiment, the media content of TIMP-1 was lowered in the PA-F + 5 µM CBG and PA-F + 10 µM CBG groups (−27.3% and −22.0%, *p* < 0.05; Figure 8C, respectively).

The protein and mRNA expressions of TIMP-1 are presented in Appendix A, Appendix A.

### 3.9. Concentration-Dependent Influence of Cannabigerol on Tissue Inhibitor of Metalloproteinase 2 Changes in Post-Incubation Media and Primary Rat Hepatocytes Exposed to Palmitate in Combination with Fructose

In the culture media, the concentration of TIMP-2 was higher after 24 h cell treatment with 1 µM CBG, and 5 µM CBG in the standard condition and treatment with 10 µM CBG in the PA-F condition (+21.9%, +5.2% and +8.5%, *p* < 0.05; Figure 9A, respectively) in comparison with the Control group. The 24 h cell treatment with 15 µM CBG with PA-F and 25 µM CBG with PA-F induced an elevation in the TIMP-2 release into media (+8.1% and +9.3%, *p* < 0.05; Figure 9B, vs. PA-F group). At the end of our experiment, we also detected an increase in the TIMP-2 concentration in post-incubation media that only occurred in the PA-F + 5µM CBG group (+1.9%, *p* < 0.05; Figure 9C) in relation to the PA-F condition.

The protein and mRNA expressions of TIMP-2 are presented in Appendix A, Appendix A.

## 4. Discussion

In the current study, we identified the effectiveness and safety concentration of cannabigerol as an agent that could be used as a protector in the development of fibrotic changes, with a low toxic effect. With regard to the LDH measurements in this study, cannabigerol at a concentration of 10 µM showed cytotoxic activity against primary hepatocytes cultured for 24 h in PA-F conditions, indicating the occurrence of membrane cell damage [20]. In addition, the cytotoxicity of cannabigerol at the 5 µM and 30 µM concentrations (after 24 h and at the end of the experiment, respectively) was significantly decreased in relation to the levels in the proper Control group, indicating a likely positive influence on cell growth and viability. A recent study reported that treatment with low concentrations of cannabigerol (1 μM and 3 μM) attenuated astrocytic LDH release in ischemic conditions, suggesting that this may be responsible for reducing cell damage [21]. This assessment of this enzyme concentration is the most widely used for a sensitive analysis of cell viability. The release of LDH into culture medium reflects the activity of cytoplasmic enzymes released by cells undergoing apoptosis or other forms of cellular damage. LDH, a stable enzyme found in all cells, is a very good marker to quantify cell cytotoxicity based on the conversion of lactate to pyruvate with NADH generation [20].

In response to chronic excessive fat accumulation, hepatocytes, as the most abundant cell type, initiate fibrosis by generating toxic pro-inflammatory and chemotactic mediators that trigger hepatocyte injury [7]. In the current investigation, we focused on the concentration-dependent effect of cannabigerol on primary rat hepatocytes, which constitute the first line in the differentiation of cells into myofibroblast and the initiation of hepatic fibrogenesis. We observed an increase in the protein expression of TGF-β1 after hepatocytes’ incubation with palmitate and fructose. In this condition, an enhancement in the release of TGF-β1 into media by the activated hepatocytes was also noted 24 h after exposure to PA-F, suggesting that fibrosis is developing. Thus, we suppose that TGF-β1 signaling may constitute a potential therapeutic target in liver fibrosis treatment. Piras et al. showed that exposure to palmitate in combination with fructose enhanced the lipotoxicity of palmitate in hepatocytes [8]. Moreover, fructose availability increases the generation of palmitic acid, accelerating lipotoxicity in the endoplasmic reticulum and increasing the intensity of the hepatic inflammatory response, which may trigger fibrogenesis [22]. In vitro studies have demonstrated that exposure to saturated fat, i.e., palmitate, in combination with fructose, significantly stimulates hepatic stellate cells’ activation, which directly synthesizes and deposits collagen, fibronectin, and other ECM components, and thus intensify fibrogenesis [8,23,24,25]. Interestingly, our results showed significant changes in the TGF-β1 protein and mRNA expression and secretion after the exposure of cells to cannabigerol. Under PA-F conditions, the most noticeable changes were observed after the incubation of hepatocytes with 10 µM CBG and 30 µM CBG, in which the TGF-β1 protein expression was decreased, while, after hepatocytes’ incubation with 25 µM CBG, the TGF-β1 protein expression was increased. Moreover, at the level of the mRNA gene, treatment with cannabigerol (10 µM CBG and 30 µM CBG) significantly decreased TGF-β1 expression, indicating a protective role in fibrogenesis by the attenuation of TGF-β1 signaling. In turn, changes in the media level of TGF-β1 indicated that 5 µM cannabigerol attenuated the release of the primary inducer of fibrosis into media from fibrotic hepatocytes at the 24 h timepoint of experimental incubation. Again, high concentrations of CBG, 10 µM CBG, and 30 µM CBG amplified the PA-F-induced secretion of TGF-β1. A previous study conducted by Aljobaily et al. reported that a high concentration of CBG accelerated the inflammation state and the deterioration of NASH induced by a methionine/choline-deficient diet, which was reflected by an increased macrophage proliferation and infiltration [10]. These facts indicate that only a low concentration of CBG can suppress the palmitate and fructose-induced TGF-β1 activation and regulate the homeostasis of ECM. In our study, treatment with PA and F resulted in a significant increase in a key marker associated with hepatic fibrosis, hydroxyproline (HYP), a major component of fibrillar-type collagen. Importantly, in all experimental groups treated with cannabigerol in combination with PA-F, hydroxyproline levels were decreased (for low concentrations of CBG) or increased (for high concentrations of CBG). The impairment in HYP noted in the current project indicates that small concentrations of this agent exclusively inhibited matrix protein accumulation and suppressed hepatic fibrosis. Research has indicated that increased HYP content is closely correlated with the stage of liver fibrosis, and may reflect histological changes in ECM composition, described as the fibrosis score [26]. The alterations in the hydroxyproline level reflect the deposition and turnover of predominant components of fibrous ECM molecules, including COL-1α1 and COL-3α1, which are secreted to a matrix and assembled in fibrils [27]. The excessive availability of PA and F upregulated COL-1α1 and COL-3α1 mRNA and protein expressions, which may account for the scar tissue formation. Previous studies reported that type I collagen overexpression promotes collagenous matrix deposition and contributes to the strengthening of connective tissue, while type 3 collagen is restricted to fibrotic processes in soft tissues such as the liver and, more importantly, correlate with the myofibroblast differentiation during hepatic fibrosis [27,28]. The present study demonstrated that cannabigerol substantially inhibited the COL-1α1 and COL-3α1 expression levels of proteins and genes in selected groups of hepatocytes upon treatment with PA-F. It can be hypothesized that exposure to low concentrations of CBG, especially at a concentration of 5 µM, had strong antifibrogenic properties, triggering the downregulation of the mRNA gene and protein expression of both examined types of collagens. In this context, decreases in collagen expressions reflected an attenuated matrix protein deposition and cell differentiation, which led to the inhibition of the development of fibrotic responses in primary hepatocytes exposed to PA-F. Moreover, cannabigerol, as an agonist of the transient receptor potential cation channel subfamily V member 1 (TRPV1), can inhibit the reuptake of anandamide (AEA). Consistent with this fact, a higher AEA concentration in circulation triggers the apoptosis of hepatic myofibroblasts, which are responsible for the excessive synthesis, deposition, and remodeling of ECM proteins, and finally limits the development of fibrotic changes [13,29,30]. Of note, the deposition and turnover of ECM compounds remain under the control of secreted enzymes, i.e., matrix metalloproteinase and their specific tissue inhibitors, TIMP [31,32]. The imbalance of MMP and TIMP underlies the pathomechanism of liver fibrosis, in which the predominance of TIMP level is observed [12,32]. The present investigation revealed that, after the incubation of hepatocytes with palmitate and fructose media, the mRNA level of gelatinases, MMP-2, and MMP-9 was markedly reduced without significant changes in their protein expression. We also observed a diminution in the secretion of MMP-2 after 24 h exposure to PA-F media and a depletion in MMP-9 level in media at the end of our experiment in the PA-F group. Matrix metalloproteinases are secreted into media and/or circulation as proenzymes and then activated in extracellular spaces, which degraded the deposition and suppressed the remodeling of ECM compounds [31,33]. Different experimental models suggest that MMP plays a crucial role in liver damage resolution and hepatic regeneration. The authors reported that MMP-2 suppresses the expression of collagen type 1 in matrix spaces and, importantly, its absence or impairment support and exacerbate fibrosis [31,34,35,36,37]. MMP-9 is involved in the release of protective factors from ECM and has a positive effect on fibrosis through the reduction in liver inflammatory processes. In addition, isoform 9 of matrix metalloproteinase promotes HSC apoptosis, improving liver failure [31,34,38,39,40]. Our investigation is consistent with these findings and suggested that cannabigerol treatment may have a protective effect on the development of fibrotic changes caused by the overexpression and/or enhanced media level of MMP-2 and MMP-9. A partially positive effect of cannabigerol on matrix metalloproteinase regulation was observed after cell exposure to a low concentration of this phytocannabinoid. In the case of MMP-2, we observed an enhancement in the protein expression (in the PA-F + 5 µM CBG and PA-F + 10 µM CBG groups), together with the secretion of MMP-2 into media at the end of the experiment (only in the PA-F + 5 µM CBG group). However, the protein expression of MMP-9 was decreased in each experimental group of hepatocytes treated with PA-F and low concentrations of CBG. Surprisingly, in the same group, the MMP-9 media level was increased after 24 h incubation and at the end of treatment, but only in the PA-F + 5µM CBG group. The enhanced generation and secretion of MMP-2 and some measure of MMP-9 after treatment with 5 µM CBG of hepatocytes exposed to palmitate and fructose solution might contribute to the reduction in experimental hepatic fibrosis. In contrast, high concentrations of CBG caused reduction in the expression of protein MMP-2 and mRNA expression of MMP-9 with a simultaneous decline in the levels of MMP-2 and MMP-9 at the end of the experiment, and an elevation in the levels of MMP-2 and MMP-9 after 24 h of exposure to post-incubation media. We suspect that CBG concentrations above 10 µM may inhibit the improvement in hepatocyte injuries and favor the acceleration of hepatic fibrosis. The balance of TIMP/MMP regulates the processes of fibrolysis and fibrogenesis. In the pathogenesis of metabolic disorders, especially hepatic fibrosis, the overexpression of TIMP was observed. The authors reported that TIMP-1 and TIMP-2 are closely related to fibrogenesis, in which enhanced TIMP-1 levels limited the activation of MMP and MMP-dependent matrix degradation and inhibited the programmed HSC death, while enhancing TIMP-2 level and inhibiting MMP-2 activity and the promotion of apoptosis [7,27,41]. According to these observations, in this study, we noted an increase in the protein expression of TIMP-1 with the enhancement of its secretion into media after 24 h incubation of hepatocytes with palmitate and fructose, without substantial changes in the expression and level of TIMP-2 in the same PA-F group. The lack of significant changes in the expression and levels of TIMP-2 may be due to the fact that this isoform of tissue MMP inhibitors is mainly produced by activated HSC, and a significant increase in its expression is observed only after prolonged exposure to factors inducing the fibrosis processes, as shown in the study conducted by Kossakowska et al. [41,42]. In the present study, cannabigerol treatment caused significant changes in TIMP determinations. Importantly, a low concentration of CBG, 5 µM, induced a reduction in the mRNA expression of TIMP-2 with a simultaneous diminishment in its secretion into media at the end of the experiment, which was consistent with the enhanced MMP-2 and/or MMP-9 protein and mRNA expression and their release into post-incubation media. Our findings suggest that CBG may activate MMP-dependent matrix degradation and restore the balance between TIMP and MMP levels, indicating the potential anti-fibrogenic effect of cannabigerol.

## 5. Conclusions

In conclusion, the data presented in this study demonstrate the concentration-dependent influence of cannabigerol on collagen deposition and the balance between TIMP and MMP via the regulation of TGF-β1 signaling in primary rat hepatocytes with fibrotic changes induced by exposure to palmitate and fructose media. Our results indicated that selected low concentrations of CBG caused a reduction in the TGF-β1 mRNA expression and secretion into media in the PA-F condition, suggesting the limitation of the TGF-β1 signaling pathway as a first line for the development of hepatic fibrosis changes. Moreover, hepatocytes’ exposure to cannabigerol at low concentrations attenuated the degradation of collagen 1 and 3 deposition in matrix space, leading to a decrease in the hydroxyproline content. The protein and/or mRNA expression and the secretion of MMP-2 and MMP-9 were augmented, which was accompanied by a diminishment in the protein and/or mRNA expression and secretion of TIMP-1 and TIMP-2. It is essential that treatment with high concentrations of CBG had adverse unfavorable effects on the liver fibrosis development. Based on these results, we suggest that cannabigerol treatment with low concentrations might expedite the regression of liver fibrosis and promote liver regeneration.

## Figures and Tables

**Figure 1 cells-12-02243-f001:**
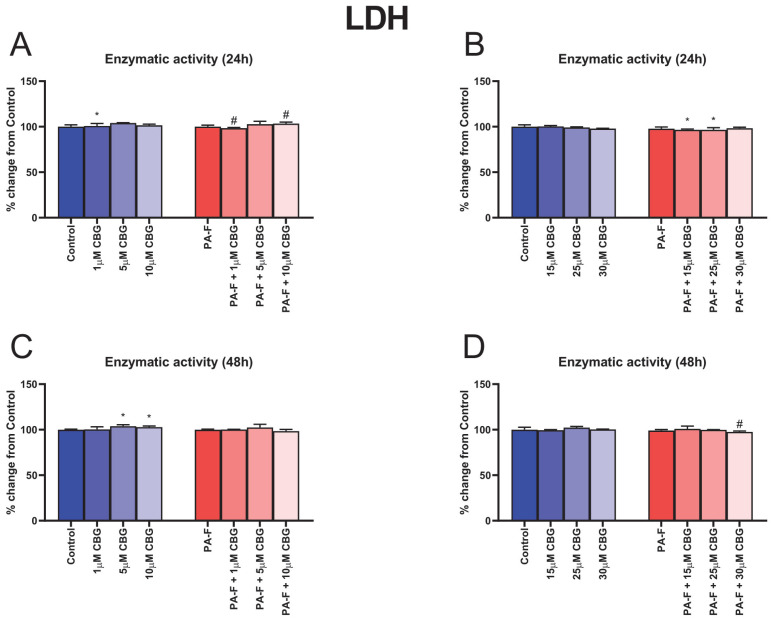
The influence of cannabigerol (CBG), in a concentration-dependent manner, on the release of lactate dehydrogenase (LDH) in culture media from hepatocytes exposed to standard media (Control) or palmitate and fructose media (PA-F). The cells were treated with different concentrations of CBG for 48 h. After 24 h of the incubation period (**A**,**B**) and at the end of the experiment, at 48 h (**C**,**D**), an LDH assay was performed using the ELISA technique to assess cell cytotoxicity level. The data are expressed as mean ± SD of six independent determinations and presented as a percentage of LDH released relative to the Control (set as 100%). Significant differences are indicated in comparison with the Control group (* *p* < 0.05) and PA-F group (^#^
*p* < 0.05).

**Figure 2 cells-12-02243-f002:**
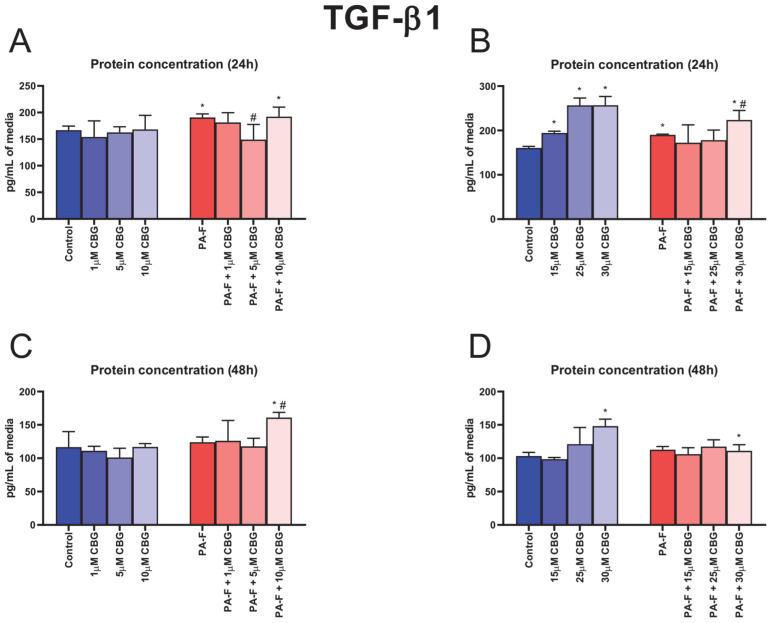
The influence of cannabigerol (CBG), in a concentration-dependent manner, on the concentration transforming growth factor beta 1 (TGF-β1) in media after 24 h incubation period (**A**,**B**) and at the end of the experiment, 48 h (**C**,**D**), from hepatocytes exposed to standard media (Control) or palmitate and fructose media (PA-F). The cells were treated with different concentrations of CBG for 48 h. The TGF-β1 concentrations in media were determined using the ELISA technique. The data are expressed as mean ± SD of six independent determinations and presented in pg/mL of media. Significant differences are indicated in comparison with the Control group (* *p* < 0.05) and PA-F group (^#^
*p* < 0.05).

**Figure 3 cells-12-02243-f003:**
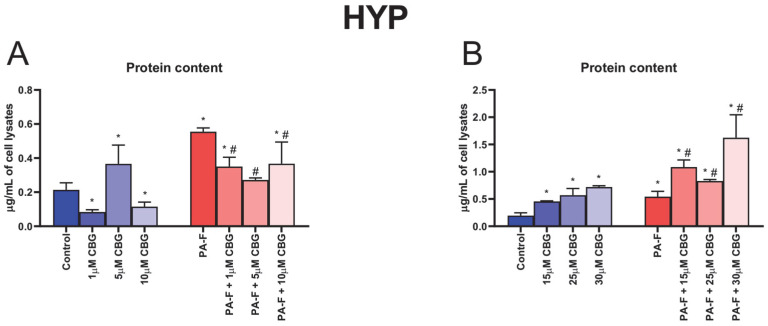
The influence of cannabigerol (CBG), in a concentration-dependent manner, on the concentration of hydroxyproline (HYP) (**A**,**B**) in the lysates of hepatocytes exposed to standard media (Control), or palmitate and fructose media (PA-F). The cells were treated with different concentrations of CBG for 48 h. The HYP assay concentration was determined using the ELISA technique. The data are expressed as mean ± SD of six independent determinations and presented in µg/mL of cell lysates. Significant differences are indicated in comparison with the Control group (* *p* < 0.05) and PA-F group (^#^
*p* < 0.05).

**Figure 4 cells-12-02243-f004:**
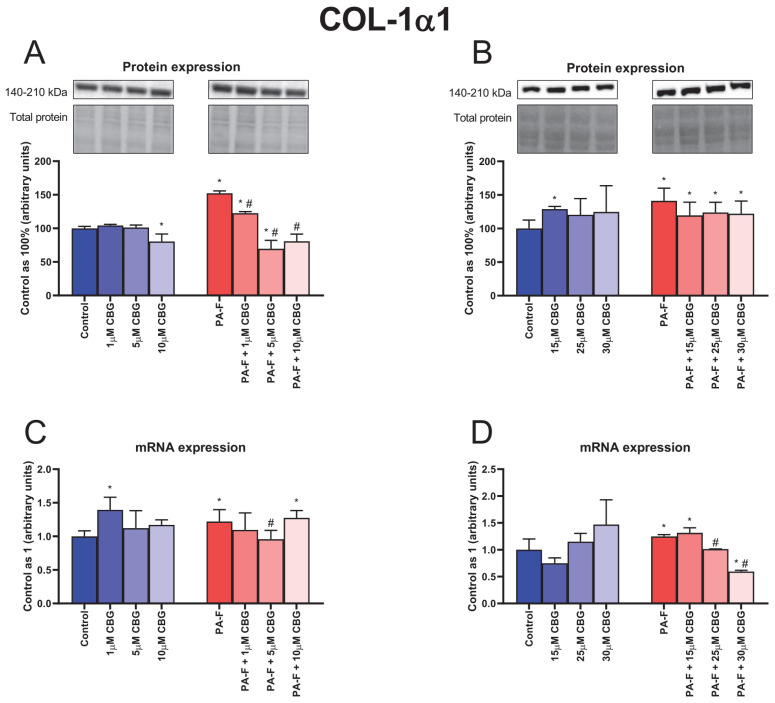
The influence of cannabigerol (CBG), in a concentration-dependent manner, on the collagen type 1 alpha 1 (COL-1α1) protein expressions (**A**,**B**), mRNA expressions (**C**,**D**) in the hepatocytes exposed to standard media (Control), or palmitate and fructose media (PA-F). The cells were treated with different concentrations of CBG for 48 h. The COL-1α1 protein and mRNA expression assays were carried out using the Western blot and RT-PCR techniques, respectively. The data are expressed as mean ± SD of six independent determinations and presented as a percentage of COL-1α1 changes relative to the Control (set as 100% for Western blot and as 1 for RT-PCR). Significant differences are indicated in comparison with the Control group (* *p* < 0.05) and PA-F group (^#^
*p* < 0.05).

**Figure 5 cells-12-02243-f005:**
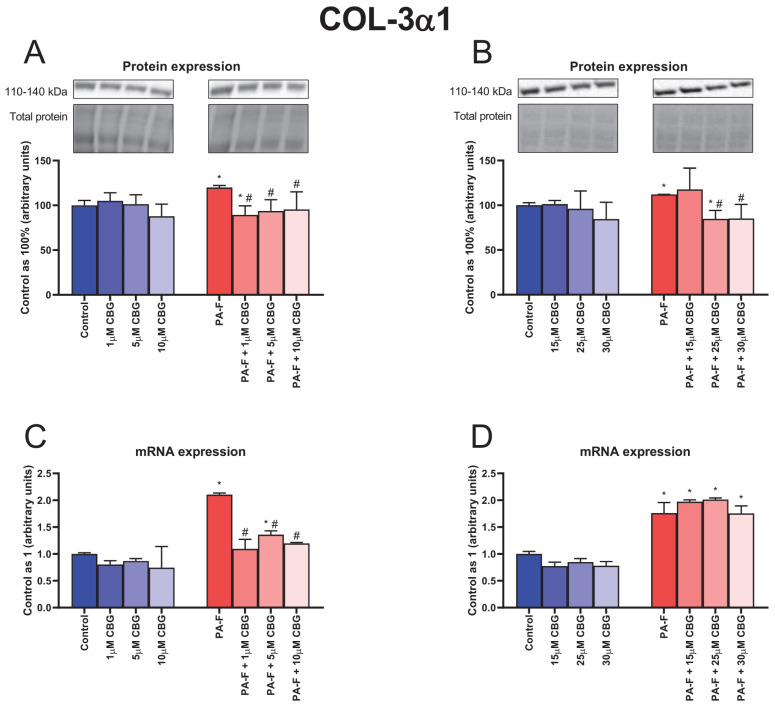
The influence of cannabigerol (CBG), in a concentration-dependent manner, on the collagen type 3 alpha 1 (COL-3α1) protein expression (**A**,**B**) and mRNA expression (**C**,**D**) in hepatocytes exposed to standard media (Control) or palmitate and fructose media (PA-F). The cells were treated with different concentrations of CBG for 48 h. The COL-3α1 protein and mRNA expression assays were performed using Western blot and RT-PCR techniques, respectively. The data are expressed as mean ± SD of six independent determinations and presented as a percentage of COL-3α1 changes relative to the Control (set as 100% for Western blot and as 1 for RT-PCR). Significant differences are indicated in comparison with the Control group (* *p* < 0.05) and PA-F group (^#^
*p* < 0.05).

**Figure 6 cells-12-02243-f006:**
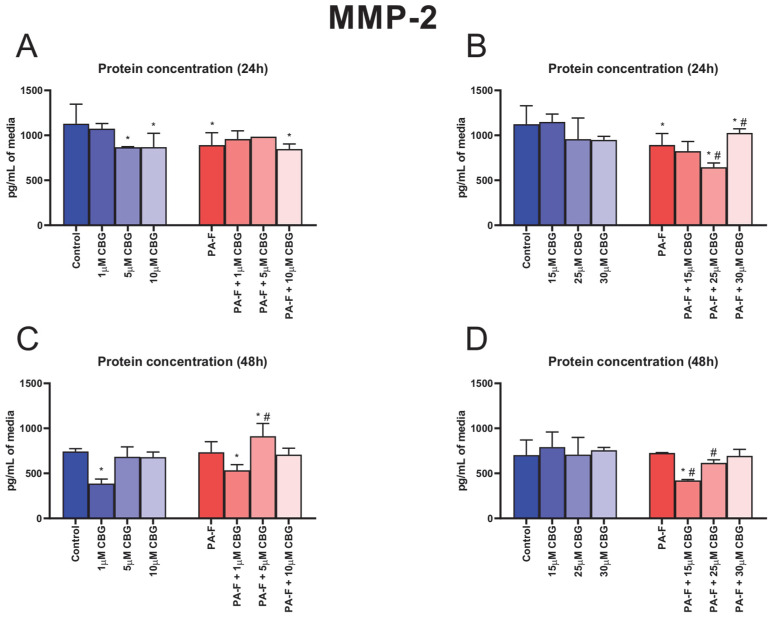
The influence of cannabigerol (CBG), in a concentration-dependent manner, on the concentration of matrix metalloproteinase 2 (MMP-2) in media after 24 h incubation period (**A**,**B**) and at the end of the experiment, 48 h (**C**,**D**), in hepatocytes exposed to standard media (Control) or palmitate and fructose media (PA-F). The cells were treated with different concentrations of CBG for 48 h. MMP-2 concentrations in media were determined using the ELISA technique. The data are expressed as the mean ± SD of six independent determinations and presented in pg/mL of media (ELISA). Significant differences are indicated in comparison with the Control group (* *p* < 0.05) and PA-F group (^#^
*p* < 0.05).

**Figure 7 cells-12-02243-f007:**
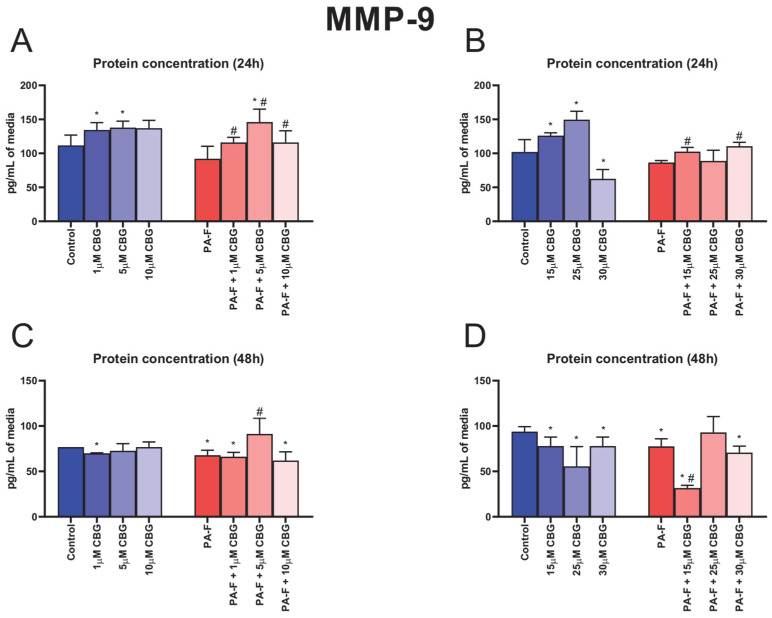
The influence of cannabigerol (CBG), in a concentration-dependent manner, on the concentration of matrix metalloproteinase 9 (MMP-9) in media after 24 h incubation period (**A**,**B**) and at the end of the experiment, 48 h (**C**,**D**), from hepatocytes exposed to standard media (Control) or palmitate and fructose media (PA-F). The cells were treated with different concentrations of CBG for 48 h. The MMP-9 concentrations were determined using the ELISA technique. The data are expressed as mean ± SD of six independent determinations and presented in pg/mL of media (ELISA). Significant differences are indicated in comparison with the Control group (* *p* < 0.05) and PA-F group (^#^
*p* < 0.05).

**Figure 8 cells-12-02243-f008:**
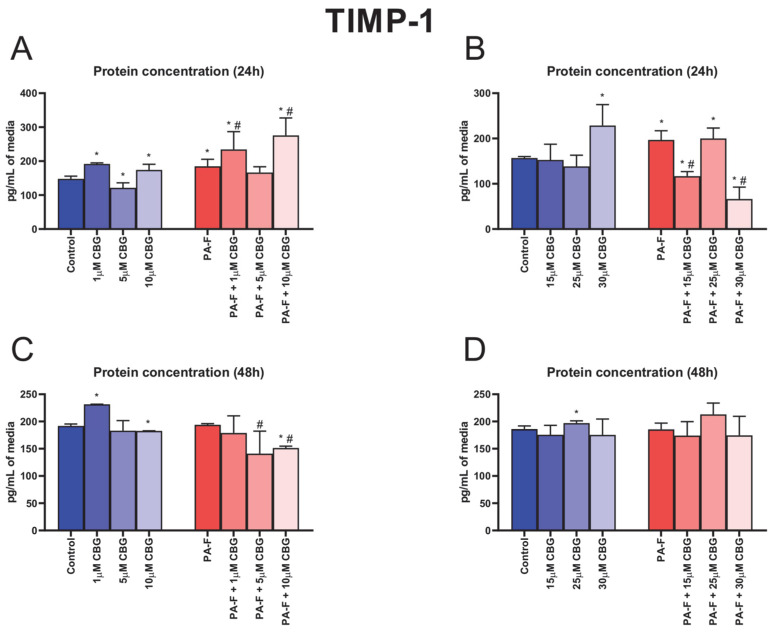
The influence of cannabigerol (CBG), in a concentration-dependent manner, on the concentration of tissue inhibitor of metalloproteinase 1 (TIMP-1) in media after 24 h incubation period (**A**,**B**) and at the end of the experiment (**C**,**D**) from hepatocytes exposed to standard media (Control) or palmitate and fructose media (PA-F). The cells were treated with different concentrations of CBG for 48 h. The TIMP-1 concentration was determined using the ELISA technique. The data are expressed as mean ± SD of six independent determinations and presented in pg/mL of media (ELISA). Significant differences are indicated in comparison with the Control group (* *p* < 0.05) and PA-F group (^#^
*p* < 0.05).

**Figure 9 cells-12-02243-f009:**
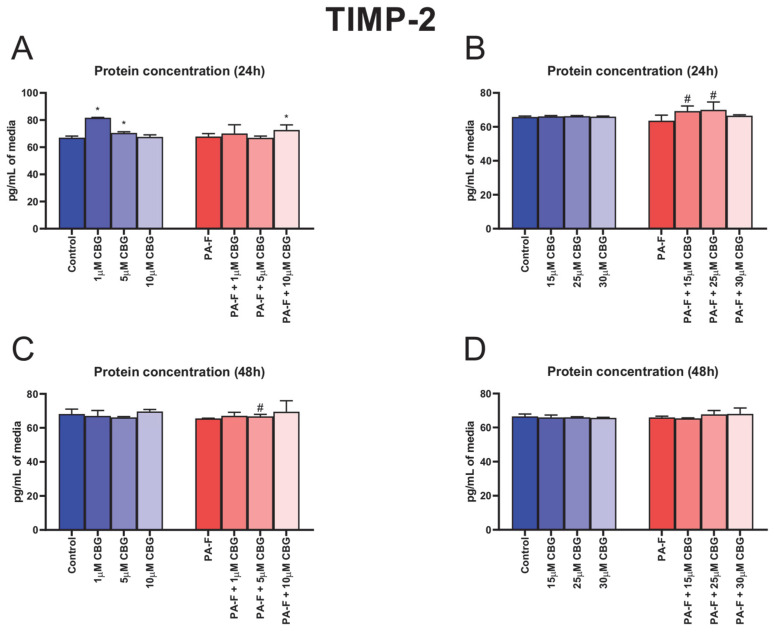
The influence of cannabigerol (CBG), in a concentration-dependent manner, on the concentration of tissue inhibitor of metalloproteinase 2 (TIMP-2) in media after 24 h incubation period (**A**,**B**) and, at the end of the experiment, 48 h (**C**,**D**), from hepatocytes exposed to standard media (Control) or palmitate and fructose media (PA-F). The cells were treated with different concentrations of CBG for 48 h. The TIMP-2 concentrations in media were determined using the ELISA technique. The data are expressed as mean ± SD of six independent determinations and presented in pg/mL of media (ELISA). Significant differences are indicated in comparison with the Control group (* *p* < 0.05) and PA-F group (^#^
*p* < 0.05).

**Table 1 cells-12-02243-t001:** Primary antibodies used for Western blot method.

Target Protein	Source (Clone Number)	Dilution	Catalog Number	Manufacturer
TGF-β1	rabbit (monoclonal)	1:200	MA5-15065	Thermo Fisher Scientific, Inc., MA, USA
MMP-2	mouse (monoclonal)	1:500	sc-13595	Santa Cruz Biotechnology, Inc., TX, USA
MMP-9	mouse (monoclonal)	1:500	sc-393859	Santa Cruz Biotechnology, Inc., TX, USA
TIMP-1	rabbit (polyclonal)	1:200	PA5-9959	Thermo Fisher Scientific, Inc., MA, USA
TIMP-2	mouse (monoclonal)	1:100	sc-365671	Santa Cruz Biotechnology, Inc., TX, USA
COL-1α1	mouse (monoclonal)	1:200	sc-293182	Santa Cruz Biotechnology, Inc., TX, USA
COL-3α1	mouse (monoclonal)	1:100	sc-514601	Santa Cruz Biotechnology, Inc., TX, USA

TGF-β1—transforming growth factor beta 1; MMP-2, -9—matrix metalloproteinase 2, 9; TIMP-1, -2—tissue inhibitor of metalloproteinase 1, 2; COL-1α1—collagen type 1 alpha 1; COL-3α1—collagen type 3 alpha 1.

**Table 2 cells-12-02243-t002:** Primer sequences used for RT-PCR assay.

Target Gene	Forward Primer Sequences (5′→3′)	Reverse Primer Sequence (5′→3′)	Annealing Temperature
TGF-β1	CTTCAGCTCCACAGAGAAGAACTGC	CACGATCATGTTGGACAACTGCTCC	64 °C
MMP-2	ACCATCGCCCATCATCAAGT	CGAGCAAAAGCATCATCCAC	60 °C
MMP-9	CTTGAAGTCTCAGAAGGTGG	AACAAGAAAGGACAGCGTGC	60 °C
TIMP-1	GGGCTAAATTCATGGGTTCC	GTTCAGGCTTCAGCTTTTGC	64 °C
TIMP-2	CCAGGTCCTTTTCATCCTGA	CTGGGACTCCTAGGCAAATG	64 °C
COL-1α1	TGGCCAAGAAGACATCCCTGAAGT	ACATCAGGTTTCCACGTCTCACCA	62 °C
COL-3α1	AGGCCAATGGCAATGTAAAG	TGTCTTGCTCCATTCACCAG	64 °C
GAPDH	TGCACCACCAACTGCTTA	GGATGCAGGGATGATGTTC	62 °C

TGF-β1—transforming growth factor beta 1; MMP-2, -9—matrix metalloproteinase 2, 9; TIMP-1, -2—tissue inhibitor of metalloproteinase 1, 2; COL-1α1—collagen type 1 alpha 1; COL-3α1—collagen type 3 alpha 1; GAPDH—glyceraldehyde-3-phosphate dehydrogenase.

## Data Availability

Data is contained within the article and Appendix A.

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
