# Peer review of "Concentration-Dependent Attenuation of Pro-Fibrotic Responses after Cannabigerol Exposure in Primary Rat Hepatocytes Cultured in Palmitate and Fructose Media"

_cells, 2023, doi:10.3390/cells12182243_

Round 1
Reviewer 1 Report
The manuscript examined a dose-dependent attenuation of pro-fibrotic responses after 2 cannabigerol exposure in primary rats hepatocytes cultured in palmitate and fructose media. According to the description, the authors identified the effectiveness and safety concentration of cannabigerol as an agent for use as a protector for the development of fibrotic changes with a low toxic effect. I consider the study as innovative and of high scrutiny to the Journal audience. I recommend publication.
Author Response
Białystok, 2022-08-23
Dear Madam, Dear Sir,
We appreciate the time and effort that you dedicated to providing your valuable feedback on our manuscript entitled "Dose-dependent attenuation of pro-fibrotic responses after cannabigerol exposure in primary rats hepatocytes cultured in palmitate and fructose media" (authors: Klaudia Sztolsztener, Karolina Konstantynowicz-Nowicka, Anna Pędzińska-Betiuk, Adrian Chabowski). We are grateful for your insightful comments on our paper. Furthermore, the whole manuscript was corrected and improved. The changes within the manuscript have been highlighted on red color and tract changes.
Here are our response to the Reviewer’s comments:
- The manuscript examined a dose-dependent attenuation of pro-fibrotic responses after 2 cannabigerol exposure in primary rats hepatocytes cultured in palmitate and fructose media. According to the description, the authors identified the effectiveness and safety concentration of cannabigerol as an agent for use as a protector for the development of fibrotic changes with a low toxic effect. I consider the study as innovative and of high scrutiny to the Journal audience. I recommend publication.
Authors: Thank you for your positive feedback.
Taking into account all the Reviewer’ suggestions and concerns provided, we believe that the manuscript was further improved and is now more suitable for publication in the Cells.
Yours faithfully,
Klaudia Sztolsztener
Department of Physiology,
Medical University of Białystok,
15-222 Białystok, Mickiewicz Str. 2 C, Poland
Email: klaudia.sztolsztener@umb.edu.pl
Telephone: +48857485585
FAX: + 48857485586

Reviewer 2 Report
In this paper Sztolsztener et al sought to examine the influence of cannabigerol (CBG) on extracellular matrix (ECM) composition in a mouse model of hepatocytes exposed to palmitate and fructose for 48h. By analysing the TGF-β1 pathway, a cornerstone in the whole hepatic fibrosis process, they reported that low-dose of CBG reduces TGF-β1 expression and attenuates collagen deposition thus exerting an anti-fibrotic action.
This is a very interesting and complex study that raise important insights in a clinically relevant field. The paper is quite well written but needs some additional edits and clarifications prior to considering for publication.
1) The authors should explain in discussion why the used the released of LDH into culture media as a cytotoxicity marker.
2) In the results (paragraph 3.1, line 274) the authors say that hepatocytes have been cultured with palmitate and fructose for 24h while in the “materials and methods” section is reported a exposure period of 48h. Which is the correct time?
3) The model is composed by two cell types: 1) Control group, which are cells cultured in standard media for 48h, and 2) Palmitic Acid with Fructose (PA-F) group, that includes cells exposed to palmitate and fructose for 48h. Half of the primary hepatocytes of both groups were incubated with CBG at different concentrations for 48h. Anyway, the authors reported the results in comparison to the control group, the PA-F group without CBG or both. In my opinion, in order to understand the real effect of CBG on hepatocytes, the two group should remain separate and PA-F cells with CBG should be compared to PA-F cell alone or with “standard” cells exposed to CBG.
4) The authors presented many data due to the complex analysis they conducted. Anyway, the text it’s too long and a bit difficult to follow. The authors could better organize results and graphs and eventually use supplementary materials.
English grammar and fluency should be partially revised. Here are reported some examples:
1) Line 48-51: “This is also accompanied by a disproportion in the expression and/or activity of regulatory enzyme families, matrix metalloproteinases (MMP) and their tissue inhibitors, tissue inhibitors of matrix metalloproteinases (TIMP), it depends ECM homeostasis [3,5].” This sentence is unclear and should be completely reformulated.
2) Line 53-56: “In general, the remodeling of ECM structure is not limited to only one specified cell type, there is a complex of liver fibrogenic cells such as hepatic stellate cells (HSC), hepatocytes, and Kupffer cells as well [2,5,8].” This sentence should become: “Different fibrogenic cell-types take part to the ECM remodelling process, such as hepatic stellate cells (HSC), hepatocytes, and Kupffer cells [2,5,8].”
3) Line 58-60: “It is a major pro-fibrogenic signal pathway, which seems to be a critical…”
4) Line 255-257: “The data from experiment were expressed as mean ± standard deviation (SD) or percentage of alteration relative to values from Control group…”
5) Line 627: it’s not “by the generation” but “by generating”.
6) Line 650: it’s not “another” but “again”
Author Response
Białystok, 2022-08-23
Dear Madam, Dear Sir,
We appreciate the time and effort that you dedicated to providing your valuable feedback on our manuscript entitled "Dose-dependent attenuation of pro-fibrotic responses after cannabigerol exposure in primary rats hepatocytes cultured in palmitate and fructose media" (authors: Klaudia Sztolsztener, Karolina Konstantynowicz-Nowicka, Anna Pędzińska-Betiuk, Adrian Chabowski). We are grateful for your insightful comments on our paper. Furthermore, the whole manuscript was corrected and improved. The changes within the manuscript have been highlighted on red color and tract changes.
Here are our response to the Reviewer’s comments:
In this paper Sztolsztener et al sought to examine the influence of cannabigerol (CBG) on extracellular matrix (ECM) composition in a mouse model of hepatocytes exposed to palmitate and fructose for 48h. By analysing the TGF-β1 pathway, a cornerstone in the whole hepatic fibrosis process, they reported that low-dose of CBG reduces TGF-β1 expression and attenuates collagen deposition thus exerting an anti-fibrotic action.
This is a very interesting and complex study that raise important insights in a clinically relevant field. The paper is quite well written but needs some additional edits and clarifications prior to considering for publication.
- The authors should explain in discussion why the used the released of LDH into culture media as a cytotoxicity marker.
Authors: We added more information about cytotoxicity marker to the Discussion section in order to clarify the use of LDH parameter.
- In the results (paragraph 3.1, line 274) the authors say that hepatocytes have been cultured with palmitate and fructose for 24h while in the “materials and methods” section is reported a exposure period of 48h. Which is the correct time?
Authors: Thank you for comment. We performed experiment where cells were incubated with palmitate and fructose, cannabigerol solutions for 48h. Importantly, experimental media were collected and replacement after 24h and 48h of treatment with PA-F and/or CBG. Thus, the measurement of selected parameter in media was performed after 24h and 48h as described in the Materials and methods section. In the given example, LDH results correctly describe the changes in media collected after 24h of incubation.
- The model is composed by two cell types: 1) Control group, which are cells cultured in standard media for 48h, and 2) Palmitic Acid with Fructose (PA-F) group, that includes cells exposed to palmitate and fructose for 48h. Half of the primary hepatocytes of both groups were incubated with CBG at different concentrations for 48h. Anyway, the authors reported the results in comparison to the control group, the PA-F group without CBG or both. In my opinion, in order to understand the real effect of CBG on hepatocytes, the two group should remain separate and PA-F cells with CBG should be compared to PA-F cell alone or with “standard” cells exposed to CBG.
Authors: In the present study, we focused on the dose-dependent CBG influence on fibrotic changes induced by PA-F. Thus, PA-F + CBG groups were compared to the Control and PA-F groups for the assessment of potential protecting CBG effect on the development of fibrosis (to determine the degree of amelioration changes comparable to the Control and PA-F groups). We did not compare PA-F + CBG groups with “standard” + CBG groups because we did not evaluate the effect of PA- F on the development of fibrosis (this is not the goal of our study; PA and F concentrations was chosen basing on availability literature).
- The authors presented many data due to the complex analysis they conducted. Anyway, the text it’s too long and a bit difficult to follow. The authors could better organize results and graphs and eventually use supplementary materials.
Authors: The results section has been precisely revised to make easier to follow and interpret. Some of the results have been moved to the Supplementary materials and the text describing the significant changes has been shortened.
- Comments on the Quality of English Language. English grammar and fluency should be partially revised. Here are reported some examples:
- Line 48-51: “This is also accompanied by a disproportion in the expression and/or activity of regulatory enzyme families, matrix metalloproteinases (MMP) and their tissue inhibitors, tissue inhibitors of matrix metalloproteinases (TIMP), it depends ECM homeostasis [3,5].” This sentence is unclear and should be completely reformulated.
- Line 53-56: “In general, the remodeling of ECM structure is not limited to only one specified cell type, there is a complex of liver fibrogenic cells such as hepatic stellate cells (HSC), hepatocytes, and Kupffer cells as well [2,5,8].” This sentence should become: “Different fibrogenic cell-types take part to the ECM remodelling process, such as hepatic stellate cells (HSC), hepatocytes, and Kupffer cells [2,5,8].”
- Line 58-60: “It is a major pro-fibrogenic signal pathway, which seems to be a critical…”
- Line 255-257: “The data from experiment were expressed as mean ± standard deviation (SD) or percentage of alteration relative to values from Control group…”
- Line 627: it’s not “by the generation” but “by generating”.
- Line 650: it’s not “another” but “again”
Authors: We appreciate spotting the language mistakes. As suggested, we clarified these throughout the manuscript.
Taking into account all the Reviewer’ suggestions and concerns provided, we believe that the manuscript was further improved and is now more suitable for publication in the Cells.
Yours faithfully,
Klaudia Sztolsztener
Department of Physiology,
Medical University of Białystok,
15-222 Białystok, Mickiewicz Str. 2 C, Poland
Email: klaudia.sztolsztener@umb.edu.pl
Telephone: +48857485585
FAX: + 48857485586

Reviewer 3 Report
MS entitled “Dose-dependent attenuation of pro-fibrotic responses after cannabigerol exposure in primary rats hepatocytes cultured in palmitate and fructose media” by Sztolsztener K et al. reports on the effect of a marijuana active component, cannabigel, on freshly isolated hepatocytes. In most of the experiments, two parameters (cannabigel concentration / fatty acid or sugar addition) were altered simultaneously. The data were rather difficult to interpret.
In ABSTRACT, the aim of the work was not clear. The authors did not state as to why it is necessary to study cannabigerol, and what it is.
In INTRODUCTION, 3 factors were introduced: TGF, marijuana component cannabigerol (CBG), and fatty acid palmitate and sugar fructose. But in terms of background information, none of these 3 were convincingly described.
Some statements in the MS seemed quite irregular to me, such as the following:
“It is accepted that hepatocytes exactly participate in the activation of TGF-β1 signaling pathway, leading to the initiation and progression of hepatic fibrosis in response to chronic stimuli [3,8].”. How would hepatocytes participate in such a process? Should it be the other way round? TGF signaling pathways in the hepatocytes have been studied?
The language used is cumbersome and unclear.
METHODS AND MATERIALS:Figure 1 is unnecessary. The collagenase digestion method of hepatocyte isolation is well established. With a cell viability of ≥ 85%, it is probably hard to interpret the LDH leakage data. Scheme 1 is unnecessary.
In RESULTS, CBG concentrations used in this work, of 5, 10, 15, 25 microM (1, 2, 3, 5-fold increase), is rather narrow and irrational (try 0.1, 0.3, 1, 3, 10, 30 .... -fold changes). The experimental schemes involved in TGF, hydroxyproline, collagens I / III, and MMP 2/9, TIMP1/2 production are rather complex and significance unclear. Two parameters were altered simultaneously, and it seemed to me that the data were rather difficult to interpret.
Needs extensive revision.
Author Response
Białystok, 2022-08-23
Dear Madam, Dear Sir,
We appreciate the time and effort that you dedicated to providing your valuable feedback on our manuscript entitled "Dose-dependent attenuation of pro-fibrotic responses after cannabigerol exposure in primary rats hepatocytes cultured in palmitate and fructose media" (authors: Klaudia Sztolsztener, Karolina Konstantynowicz-Nowicka, Anna Pędzińska-Betiuk, Adrian Chabowski). We are grateful for your insightful comments on our paper. Furthermore, the whole manuscript was corrected and improved. The changes within the manuscript have been highlighted on red color and tract changes.
Here are our response to the Reviewer’s comments:
MS entitled “Dose-dependent attenuation of pro-fibrotic responses after cannabigerol exposure in primary rats hepatocytes cultured in palmitate and fructose media” by Sztolsztener K et al. reports on the effect of a marijuana active component, cannabigel, on freshly isolated hepatocytes. In most of the experiments, two parameters (cannabigel concentration / fatty acid or sugar addition) were altered simultaneously. The data were rather difficult to interpret.
- In ABSTRACT, the aim of the work was not clear. The authors did not state as to why it is necessary to study cannabigerol, and what it is.
Authors: We understand that Abstract does not include all relevant information and therefore may seem to be not precise. We added more information about study goal and CBG; however the total amount of Abstract words should be about 200 as recommended by the Journal.
- In INTRODUCTION, 3 factors were introduced: TGF, marijuana component cannabigerol (CBG), and fatty acid palmitate and sugar fructose. But in terms of background information, none of these 3 were convincingly described.
Authors: As you suggested, we added more information to the Instruction and Discussion sections.
- Some statements in the MS seemed quite irregular to me, such as the following:
“It is accepted that hepatocytes exactly participate in the activation of TGF-β1 signaling pathway, leading to the initiation and progression of hepatic fibrosis in response to chronic stimuli [3,8].”. How would hepatocytes participate in such a process? Should it be the other way round? TGF signaling pathways in the hepatocytes have been studied?
Authors: Thank you for this comment. We moderated this sentence. In the herein study, we focused on the effect of cannabigerol (including changes in the expression of gene and protein and media concentration of TGF-β1) without examination this signaling pathway. In the future our data will be expanded, including the mechanism of these changes, in hepatocytes and other types of cell during fibrotic processes.
- The language used is cumbersome and unclear.
Authors: The manuscript undergone extensive language editing.
- METHODS AND MATERIALS:Figure 1 is unnecessary. The collagenase digestion method of hepatocyte isolation is well established. With a cell viability of ≥ 85%, it is probably hard to interpret the LDH leakage data. Scheme 1 is unnecessary.
Authors: As you suggested, we delated the Figure 1 and Scheme 1.
- In RESULTS, CBG concentrations used in this work, of 5, 10, 15, 25 microM (1, 2, 3, 5-fold increase), is rather narrow and irrational (try 0.1, 0.3, 1, 3, 10, 30 .... -fold changes). The experimental schemes involved in TGF, hydroxyproline, collagens I / III, and MMP 2/9, TIMP1/2 production are rather complex and significance unclear. Two parameters were altered simultaneously, and it seemed to me that the data were rather difficult to interpret.
Authors: The concentration of cannabigerol solution was selected based on available literature, taking into account the limits of negative impact on cell viability. In the herein study, we focused on cannabigerol impact (at concentration 1-30 µM) on hepatic fibrosis. In the future, in order to get to know more about CBG properties we will determine how a higher (to 30-fold) concentration of CBG impacts on the fibrotic processes in hepatocytes. We added this information into Materials and Methods section with the references).
The examined parameters were determined at the mRNA gene and protein level as well as the content in post-incubation media, therefore, these affects the complexity of the Result section and Figures presented in this part of manuscript. Moreover, the results section has been precisely revised to make easier to follow and interpret. Some of the results have been moved to the Supplementary materials and the text describing the significant changes has been shortened.
- Comments on the Quality of English Language
Needs extensive revision.
Authors: The manuscript undergone extensive language editing.
Taking into account all the Reviewer’ suggestions and concerns provided, we believe that the manuscript was further improved and is now more suitable for publication in the Cells.
Yours faithfully,
Klaudia Sztolsztener
Department of Physiology,
Medical University of Białystok,
15-222 Białystok, Mickiewicz Str. 2 C, Poland
Email: klaudia.sztolsztener@umb.edu.pl
Telephone: +48857485585
FAX: + 48857485586

Round 2
Reviewer 3 Report
MS basically describes a half dose response (1, 10, 30 microM) relationship and less than half time course (24, 48 hrs). The work is basically incomplete.
When intending to do a dose response relationship, do a proper dose relationship (gradients of 0.1, 0.3, 1, 3, 10, 30, 100 ... ). Plot only the useful doses (1, 10, 30 microM). A lot of useless doses were used in the present work, which makes the work tedious and un-interesting. Authors should read some basic pharmacology, especially the doses that should be chosen to use.
When intending to do a time course, do a proper time course, not just 2 time points (24, 48 hrs). Why is it necessary to repeat the two time points for the expression of each and every parameter?
English remains poor.
Author Response
Białystok, 2023-09-06
Dear Madam, Dear Sir,
We appreciate the time and effort that you dedicated to providing your valuable feedback on our manuscript entitled "Dose-dependent attenuation of pro-fibrotic responses after cannabigerol exposure in primary rats hepatocytes cultured in palmitate and fructose media" (authors: Klaudia Sztolsztener, Karolina Konstantynowicz-Nowicka, Anna Pędzińska-Betiuk, Adrian Chabowski). We are grateful for your insightful comments on our paper. Furthermore, the whole manuscript was corrected and improved. The changes within the manuscript have been highlighted and tract changes.
MS basically describes a half dose response (1, 10, 30 microM) relationship and less than half time course (24, 48 hrs). The work is basically incomplete.
When intending to do a dose response relationship, do a proper dose relationship (gradients of 0.1, 0.3, 1, 3, 10, 30, 100 ... ). Plot only the useful doses (1, 10, 30 microM). A lot of useless doses were used in the present work, which makes the work tedious and un-interesting. Authors should read some basic pharmacology, especially the doses that should be chosen to use.
When intending to do a time course, do a proper time course, not just 2 time points (24, 48 hrs). Why is it necessary to repeat the two time points for the expression of each and every parameter?
Authors: Thank you for this comment. We conducted our experiment based on previously study, which was cited in this manuscript. We choose the same range of CBG concentration (1-30 µM) from the cited paper; however, we added a few additional concentrations ourselves, so as not to copy the cited studies. Importantly, our research was conducted on a different cell model, which is associated with different expression of CBG receptors. As suggested by the reviewer, our determinations of selected parameters expression/content in the experimental medium were performed both after 24h and at the end of the experiment (48h).
- English remains poor.
Authors: Our manuscript undergone extensive language editing by MDPI English Editing service, which has been confirmed by a certificate.
Taking into account all suggestions and concerns provided, we believe that the manuscript was further improved and is now more suitable for publication in the Cells.
Yours faithfully,
Klaudia Sztolsztener
Department of Physiology,
Medical University of Białystok,
15-222 Białystok, Mickiewicz Str. 2 C, Poland
Email: klaudia.sztolsztener@umb.edu.pl
Telephone: +48857485585
FAX: + 48857485586
